# Substantial near-infrared radiation-driven photosynthesis of chlorophyll *f*-containing cyanobacteria in a natural habitat

Michael Kühl[1,2]*, Erik Trampe[1], Maria Mosshammer[1], Michael Johnson[3], Anthony WD Larkum[2], Niels-Ulrik Frigaard[1], Klaus Koren[1,4]

[1]Marine Biological Section, Department of Biology, University of Copenhagen, Copenhagen, Denmark; [2]Climate Change Cluster, University of Technology Sydney, Sydney, Australia; [3]iThree Institute, University of Technology Sydney, Sydney, Australia; [4]Centre for Water Technology, Section for Microbiology, Department of Bioscience, University of Aarhus, Aarhus, Denmark

**Abstract** Far-red absorbing chlorophylls are constitutively present as chlorophyll (Chl) *d* in the cyanobacterium *Acaryochloris marina*, or dynamically expressed by synthesis of Chl *f*, red-shifted phycobiliproteins and minor amounts of Chl *d* via far-red light photoacclimation in a range of cyanobacteria, which enables them to use near-infrared-radiation (NIR) for oxygenic photosynthesis. While the biochemistry and molecular physiology of Chl *f*-containing cyanobacteria has been unraveled in culture studies, their ecological significance remains unexplored and no data on their in situ activity exist. With a novel combination of hyperspectral imaging, confocal laser scanning microscopy, and nanoparticle-based $O_2$ imaging, we demonstrate substantial NIR-driven oxygenic photosynthesis by endolithic, Chl *f*-containing cyanobacteria within natural beachrock biofilms that are widespread on (sub)tropical coastlines. This indicates an important role of NIR-driven oxygenic photosynthesis in primary production of endolithic and other shaded habitats.

*For correspondence: mkuhl@bio.ku.dk

**Competing interests:** The authors declare that no competing interests exist.

## Introduction

The persisting textbook notion that oxygenic photosynthesis is mainly driven by visible wavelengths of light (400–700 nm) and chlorophyll (Chl) *a* as the major photopigment is challenged; recent findings indicate that cyanobacteria with red-shifted chlorophylls and phycobiliproteins capable of harvesting near-infrared-radiation (NIR) at wavelengths > 700–760 nm and exhibiting a pronounced plasticity in their photoacclimatory responses (*Gan et al., 2014*; *Gan and Bryant, 2015*) are widespread in natural habitats (*Gan et al., 2015*; *Zhang et al., 2019*; *Behrendt et al., 2019*). Besides the Chl *d*-containing cyanobacterium *Acaryochloris marina*, which was originally isolated from tropical ascidians (*Miyashita, 2014*) but has now been found in many other habitats (*Behrendt et al., 2011*; *Zhang et al., 2019*), the discovery of Chl *f* (*Chen et al., 2010*) and its occurrence in many different cyanobacteria (*Gan et al., 2015*) has triggered a substantial amount of research on the biochemistry and molecular physiology of Chl *f*-containing cyanobacterial strains (*Airs et al., 2014*; *Allakhverdiev et al., 2016*; *Chen, 2014*; *Ho et al., 2016*; *Nürnberg et al., 2018*). In comparison, the in situ distribution and activity of Chl *f*-containing cyanobacteria and their role in primary productivity remain largely unexplored. Here, we used a novel combination of hyperspectral imaging, confocal laser scanning microscopy, and chemical imaging of $O_2$ for high-resolution mapping of the distribution of Chl *f*–containing cyanobacteria and their NIR-driven oxygenic photosynthesis in an intertidal beachrock habitat. Our study gives novel insight into the ecological niche and importance of endolithic Chl *f*-containing cyanobacteria, and indicates that high rates of NIR-driven oxygenic photosynthesis can contribute to primary production in natural biofilm habitats.

With an in vivo absorption range of 700–760 nm, Chl *f* is the most red-shifted chlorophyll, which was first found in the filamentous cyanobacterium *Halomicronema hongdechloris* isolated from stromatolites in Western Australia (*Chen et al., 2010*; *Chen et al., 2012*), and in an unicellular cyanobacterium (strain KC1 related to *Aphanocapsa* spp.) isolated from Lake Biwa, Japan (*Akutsu et al., 2011*). However, Bryant and coworkers discovered that the ability to synthesize Chl *f*, far-red shifted phycobiliproteins, and small amounts of Chl *d* can be induced in many different cyanobacteria, including representatives from all five major subdivisons, when grown under far-red light-enriched conditions (*Gan et al., 2014*; *Gan et al., 2015*). Such far-red light photoacclimation (FaRLiP) involves remodeling of the photosynthetic apparatus via synthesis and modification of pigments and pigment-protein complexes. This remodeling is primarily regulated at the transcriptional level via upregulation of paralogous photosynthesis-related genes in a 21-gene cluster, which seems largely conserved in cyanobacteria exhibiting FaRLiP (*Gan and Bryant, 2015*) and contains genes for a red phytochrome-triggered control cascade of FaRLiP (*Zhao et al., 2015*). Recently, it was shown that FaRLiP also involves the modification and inclusion of Chl *f* in both PSI and PSII (*Itoh et al., 2015*; *Nürnberg et al., 2018*).

Besides their strong relevance for exploring the biophysical and biochemical limits and controls of oxygenic photosynthesis, cyanobacteria with Chl *d* and Chl *f* have been employed in speculations on more efficient light harvesting of solar energy, as organisms with red-shifted photopigments can in principle exploit ~20% more photons in the solar spectrum for their photosynthesis (*Chen and Blankenship, 2011*). However, FaRLiP involves remodeling of the photosynthetic apparatus for better performance under 700–760 nm light and is apparently not induced in absence of NIR (*Gan et al., 2014*), and *Acaryochloris marina* has largely exchanged its Chl *a* with Chl *d* (*Miyashita, 2014*). Hence, the selective benefit of employing Chl *d*, Chl *f*, and red-shifted phycobiliproteins in oxygenic photosynthesis may be more related to the capability of exploring special ecological niches in the shadow of other oxygenic phototrophs (*Ohkubo and Miyashita, 2017*; *Kühl et al., 2005*). But we know nothing about the quantitative role of cyanobacteria with far red-shifted photopigments for primary production in natural habitats.

There is a growing database of cyanobacteria and habitats wherein Chl *f* has been detected (*Supplementary file 1*). However, only three studies have reported on the actual distribution and niche of Chl *f*-containing cyanobacteria in their natural habitats (*Trampe and Kühl, 2016*; *Ohkubo and Miyashita, 2017*; *Behrendt et al., 2019*), and hitherto NIR-driven oxygenic photosynthesis by Chl *f*-containing cyanobacteria has not been demonstrated in situ. This is experimentally challenging, as cyanobacteria exhibiting FaRLiP respond to the local light microenvironment and typically occur in dense proximity with other oxygenic phototrophs harboring a plethora of photopigments fueling Chl *a*-based oxygenic photosynthesis by visible light (400–700 nm) (*Ohkubo and Miyashita, 2017*; *Behrendt et al., 2015*). Consequently, photosynthetic activity of Chl *f*-containing cyanobacteria has only been measured on strains (e.g. *Gan et al., 2014*; *Nürnberg et al., 2018*) or enrichments isolated from their natural habitat (e.g. *Behrendt et al., 2015*). Detailed microscopic investigation of pigmentation in cyanobacteria with Chl *f* has also largely been limited to culture material (*Majumder et al., 2017*; *Zhang et al., 2019*).

Beachrock is a widespread sedimentary rock formation on (sub)tropical, intertidal shorelines, where a mixture of biogeochemical processes cement carbonate sands together into a porous solid matrix (*Vousdoukas et al., 2007*). The upper surface of light-exposed beachrock is colonized by dense microbial biofilms dominated by cyanobacteria (*Cribb, 1966*; *Díez et al., 2007*), which are embedded in a dense exopolymeric matrix covering the surface and endolithic pore space of the beachrock (*Petrou et al., 2014*). We previously demonstrated the presence of Chl *f*-containing cyanobacteria in an endolithic niche below the surface biofilm along with the ability of beachrock samples to upregulate their Chl *f*-content upon incubation under NIR, indicative of FaRLiP (*Trampe and Kühl, 2016*). Here, we correlate the spatial organization of Chl *f*-containing cyanobacteria with direct in vivo measurements of their NIR-driven $O_2$ production in a natural beachrock habitat.

## Results and discussion

Hyperspectral reflectance imaging on vertical cross-sections of beachrock submerged in seawater (23°C and salinity = 35) revealed the presence of a dense ~1 mm thick surface biofilm with high amounts of Chl *a*, while a more patchy zone containing Chl *f*, and less Chl *a* was found below the

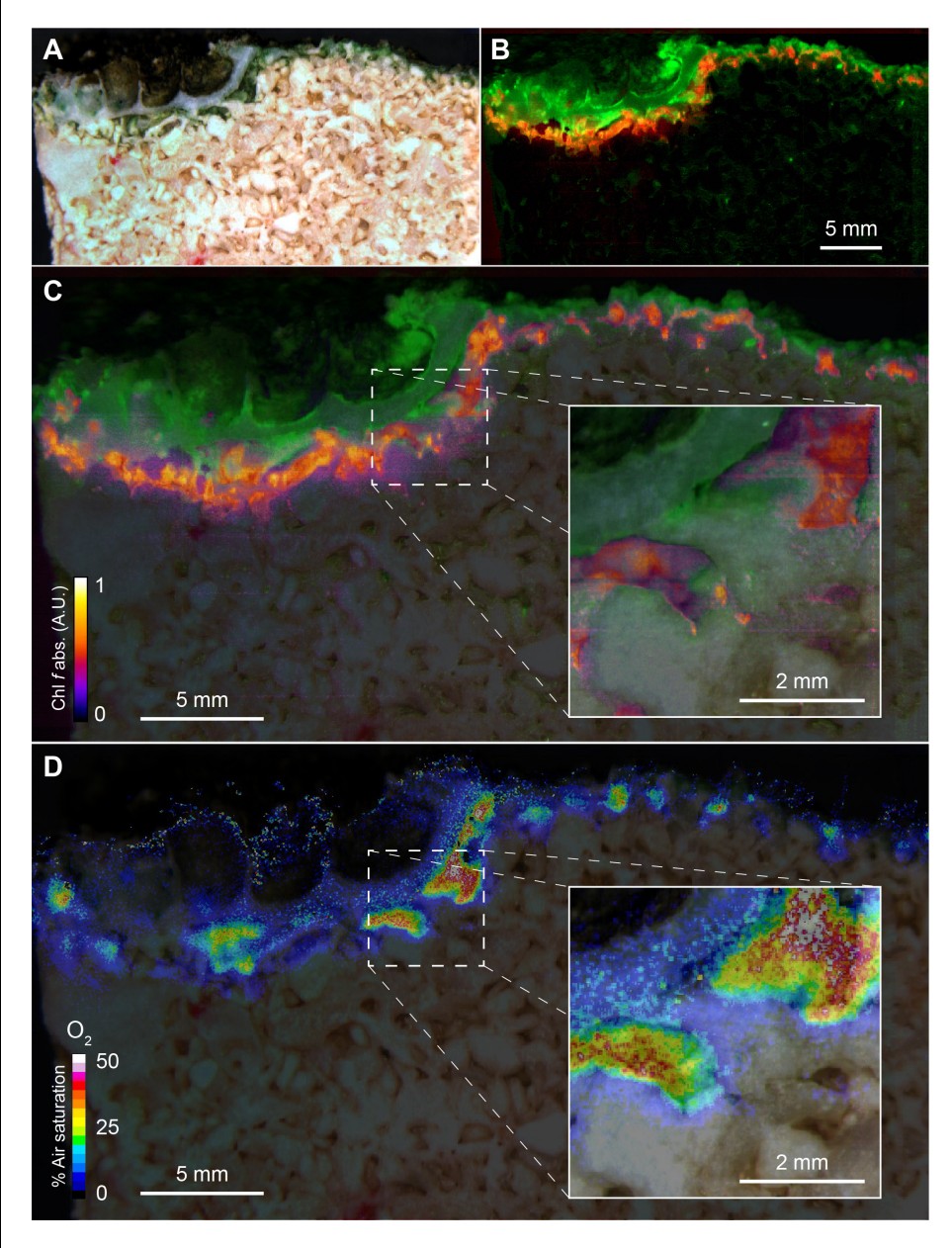

**Figure 1.** Spatial distribution of photopigments and near-infrared radiation-driven oxygenic photosynthesis in beachrock as mapped with hyperspectral reflectance imaging and chemical imaging of $O_2$. (**A**) RGB image composite, constructed from the hyperspectral image stack (R = 650 nm, G = 550 nm, B = 450 nm), showing 'true' colors of beachrock material and the biofilm community in a cross-section of the top layer. (**B**) False color coded image of the same hyperspectral image stack as in panel A mapping pixels with Chl *a* absorption (670–680 nm) in green, and Chl *f* absorption (718–722 nm) in red. Representative reflectance spectra of the two regions are given in *Figure 1—figure supplement 1*. (**C**) Overlay of beachrock structure obtained in panel A and the Chl *a* signature from panel B with map of the relative abundance of Chl *f* obtained from the amplitude of Chl *f* absorption (color coded between 0 and 1), as acquired from hyperspectral image analysis. (**D**) Distribution of $O_2$ concentration (color coded in units of % air saturation) in the beachrock under illumination of 740 nm light (half-bandwidth = 25 nm; photon irradiance = 28 µmol photons $m^{-2}$ $s^{-1}$) when immersed in anoxic seawater, as imaged with the beachrock section covered with a thin paint of agarose containing $O_2$–sensitive nanoparticles. The $O_2$ concentration image was superimposed onto the structural image of the beachrock cross section. The insert is a digital zoom corresponding to the insert in panel C. Additional data on two other beachrock sections are available in the Suppl. Materials (*Figure 1—figure supplements 2* and *6*).

*Figure 1 continued on next page*

*Figure 1 continued*

The online version of this article includes the following figure supplement(s) for figure 1:

**Figure supplement 1.** Reflectance spectra of beachrock.
**Figure supplement 2.** Macroscopic distribution of Chl *f* in beachrock.
**Figure supplement 3.** Pigment analysis of black beachrock.
**Figure supplement 4.** Confocal laser scanning microscopy (CLSM) of a beachrock cross-sectional area.
**Figure supplement 5.** Higher resolution CLSM of beachrock.
**Figure supplement 6.** Application of agarose paint with nanoparticles.
**Figure supplement 7.** NIR-driven $O_2$ production by endolithic cyanobacteria in beachrock.
**Figure supplement 8.** Calibration curve of the nanoparticle-based $O_2$ sensor paint.

surface biofilm of the beachrock (*Figure 1A,B*), exhibiting localized hot spots of Chl *f* concentration (*Figure 1C*). Representative reflectance spectra from these regions carrying spectral signatures of maximal Chl *a* and Chl *f* absorption at 670–680 nm and 715–725 nm, respectively, are presented in the Supplementary Materials (*Figure 1—figure supplement 1*) along with additional examples of hyperspectral imaging of beachrock cross-sections (*Figure 1—figure supplement 2*). The presence of Chl *f* (and absence of microalgal Chl *b* and *c*) in our samples was confirmed by HPLC analysis of beachrock pigment extracts (see Materials and methods), both in a ~ 0–2 mm thick black beachrock sample, and in a more distinct green endolithic layer (~2–5 mm below the beachrock surface) (*Figure 1—figure supplement 3*). The amount of Chl *f* relative to Chl *a* in these samples ranged from 3.5% in the mixed layer to 6% in the green layer, while *Trampe and Kühl (2016)* reported Chl *f* amounts ranging from 0.5% to 5% of Chl *a* in beachrock.

To further describe the microscale distribution of cells with different photo-pigmentation, we employed hyperspectral fluorescence imaging with confocal laser scanning microscopy (CLSM; 488 nm excitation) on beachrock cross-sections (*Figure 1—figure supplements 4* and *5*). The CLSM data confirmed the occurrence of patches of filamentous and unicellular Chl *f*-containing cyanobacteria with a characteristic fluorescence peak around 740–750 nm (cf. *Majumder et al., 2017*) in deeper endolithic zones (*Figure 1—figure supplements 4C* and *5A–D*). Besides filamentous morphotypes, brightfield microscopy of Chl *f* hot spots revealed the presence of larger round cell aggregates (*Figure 1—figure supplement 5E,F*) typical of pleurocapsalean cyanobacteria (*Waterbury and Stanier, 1978*).

In order to confirm the apparent dominance of cyanobacteria over microalgal oxygenic phototrophs, we employed 16S rRNA gene amplicon sequencing on black beachrock samples taken from the same area as the samples used for hyperspectral and $O_2$ imaging. Among ~39,000 cyanobacterial-like sequences obtained from the black beachrock, none were classified as chloroplasts. In contrast, among ~17,000 cyanobacterial-like sequences obtained from two samples of seawater, 18% were classified as chloroplasts (*Supplementary file 2*). Microalgae are thus likely to be completely absent from the black beachrock, where the oxygenic phototrophic community consists exclusively of cyanobacteria.

Analysis of a distinct green layer below the black beachrock surface biofilm showed that most cyanobacterial OTU's were clustering with *Halomicronema* (harboring the Chl *f*-containing species *Halomicronema hongdechloris*; *Chen et al., 2012*) as well as coccoid *Pseudocapsa* and *Chroococcidiopsis* (*Supplementary file 2*). The surface layer of the black beachrock harbored a higher cyanobacterial diversity with most OTU's clustering with *Rivularia*, *Calothrix*, and *Halomicronema*, as well as numerous smaller populations clustering with *Chroococcidiopsis* and other coccoid cyanobacteria. These data confirm earlier findings of i) Chl *f* (and only minor amounts of Chl *d*) in beachrock (*Trampe and Kühl, 2016*), and ii) a predominance of cyanobacteria as the major oxygenic phototrophs in beachrock (*Cribb, 1966*; *Díez et al., 2007*). We note that a comprehensive description of the cyanobacterial diversity associated with beachrock was beyond the scope of the present study, and a detailed study of the microbial diversity in beachrock (based on 16S rRNA amplicon and metagenomic sequencing) will be presented elsewhere. Here, we focus on the photosynthetic activity of Chl *f*-containing cyanobacteria in beachrock.

By coating beachrock cross-sections with a thin (<1 mm) layer of an $O_2$-sensitive nanoparticle-agarose paint (see Materials and methods and *Figure 1—figure supplement 6*) and subsequent immersion in anoxic water, it was possible to map the local $O_2$ production over the beachrock cross-

section when illuminated with weak NIR levels (740 nm, 25 nm half bandwidth; 28 μmol photons $m^{-2}$ $s^{-1}$). We observed hot spots of NIR-driven photosynthesis driving local $O_2$ levels from 0% to >40–50% air saturation within 15–20 min (*Figure 1D*, *Figure 1—figure supplement 7*), which overlapped with regions of high Chl *f* absorption (*Figure 1C*, *Figure 1—figure supplement 3*). The build-up of $O_2$ in the hotspots harboring Chl *f* occurred rapidly after onset of NIR illumination and dissipated rapidly back to anoxia within a few minutes after darkening (see *Video 1*). Based on $O_2$ concentration images recorded at 5 min intervals after experimental light-dark shifts, we calculated images of apparent dark respiration and NIR-driven net and gross photosynthesis that could be mapped onto the beachrock structure (*Figure 2A–D*) showing that hotspots of activity aligned with the presence of Chl *f* (see *Figure 1B,C*). We extracted estimates of maximum $O_2$ conversion rates in particular regions of interest (ROI) showing high rates of NIR-driven gross photosynthesis of ~5–15 μmol $O_2$ $L^{-1}$ $min^{-1}$ in the beachrock under the given actinic irradiance of 28 μmol photons $m^{-2}$ $s^{-1}$; a similar range was found for two other beachrock cross-sections (data not shown). These volume-specific rates fall among the upper range of maximal gross photosynthesis rates in aquatic phototrophs, and are comparable with photosynthetic rates found in benthic microalgae (*Krause-Jensen and Sand-Jensen, 1998*).

The Chl *f*-driven oxygenic photosynthesis by endolithic cyanobacteria thus seems very efficient given the low actinic NIR level applied. High photosynthetic efficiency of Chl *f*-containing cyanobacteria under NIR has been shown in previous *ex situ* studies on cultivated strains and enrichments. *Gan et al. (2014)* thus found that *Leptolyngbya* strain JSC-1 cells showed a 40% higher $O_2$ production rate with NIR after undergoing FaRLiP relative to cells grown under red light (*Gan et al., 2014*), and a similar high photosynthetic efficiency was found in a *Chroococcidiopsis* strain (*Nürnberg et al., 2018*). Using similar NIR levels as used in the present study, *Behrendt et al. (2015)* showed rapid saturation of NIR-driven oxygenic photosynthesis already at 25–30 μmol photons $m^{-2}$ $s^{-1}$ (740 nm) in a cell enrichment with Chl *f*-containing, *Aphanocapsa*-like cyanobacteria from a cavernous biofilm.

Our structural and chemical imaging of beachrock showed that Chl *f* and NIR-driven $O_2$ production was confined to a relative narrow zone 1–2 mm below the beachrock surface in the investigated samples (*Figure 1*, *Figure 2*; *Figure 1—figure supplements 2*, *4* and *5*). Assuming a NIR-driven oxygenic photosynthesis rate of 10 μmol $O_2$ $L^{-1}$ $min^{-1}$ (=nmol $O_2$ $cm^{-3}$ beachrock $min^{-1}$) in a 1 mm thick layer (*Figure 2*), and a conservative estimate of beachrock porosity of ~0.4 in the uppermost 1–2 mm (*Vousdoukas et al., 2007*), we can estimate the areal NIR-driven gross photosynthesis rate to (10 μmol $L^{-1}$ $min^{-1}$ × 0.4×0.1 cm x $10^{-6}$ x 10000 x 60 =) ~0.24 mmol $O_2$ $m^{-2}$ beachrock $h^{-1}$. Total beachrock primary productivity remains to be quantified in detail, but it is well known that beachrock habitats sustain high grazing rates of epifauna (*McLean, 1974*; *McLean, 2011*) and herbivorous reef fish (*Stephenson and Searles, 1960*).

To our knowledge, beachrock primary production has only been reported by Krumbein, who studied a Red Sea beachrock habitat (*Krumbein, 1979*). Based on his data on $O_2$ exchange under water covered conditions (cf. Figure 13 in *Krumbein, 1979*), we estimated a total gross photosynthesis of ~10 mmol $O_2$ $m^{-2}$ beachrock $h^{-1}$. While based on several crude assumptions and comparing different habitats, these rough calculations indicate that NIR-driven photosynthesis could account for at least 2–3% of total areal photosynthesis in beachrock habitats. Furthermore, we note that we base this estimate on $O_2$ dynamics over minutes measured

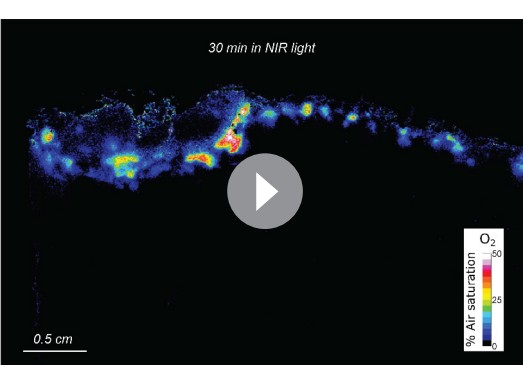

30 min in NIR light

0.5 cm

**Video 1.** Animation of NIR-driven $O_2$ dynamics over a beachrock cross-section (see *Figures 1* and *2*) coated with a thin (<1 mm) agarose layer with luminescent $O_2$ sensor nanoparticles. The movie sequence shows the decline in $O_2$ concentration (recorded at 5 min interval) starting from steady-state conditions under a NIR irradiance (740 nm; 25 nm HBW) of 28 μmol photons $m^{-2}$ $s^{-1}$ approaching steady-state dark conditions after 35 min, followed by the rise in $O_2$ concentration over 25 min after switching the NIR irradiation on again. The colored scale bar relates the colors to $O_2$ concentrations.

https://elifesciences.org/articles/50871#video1

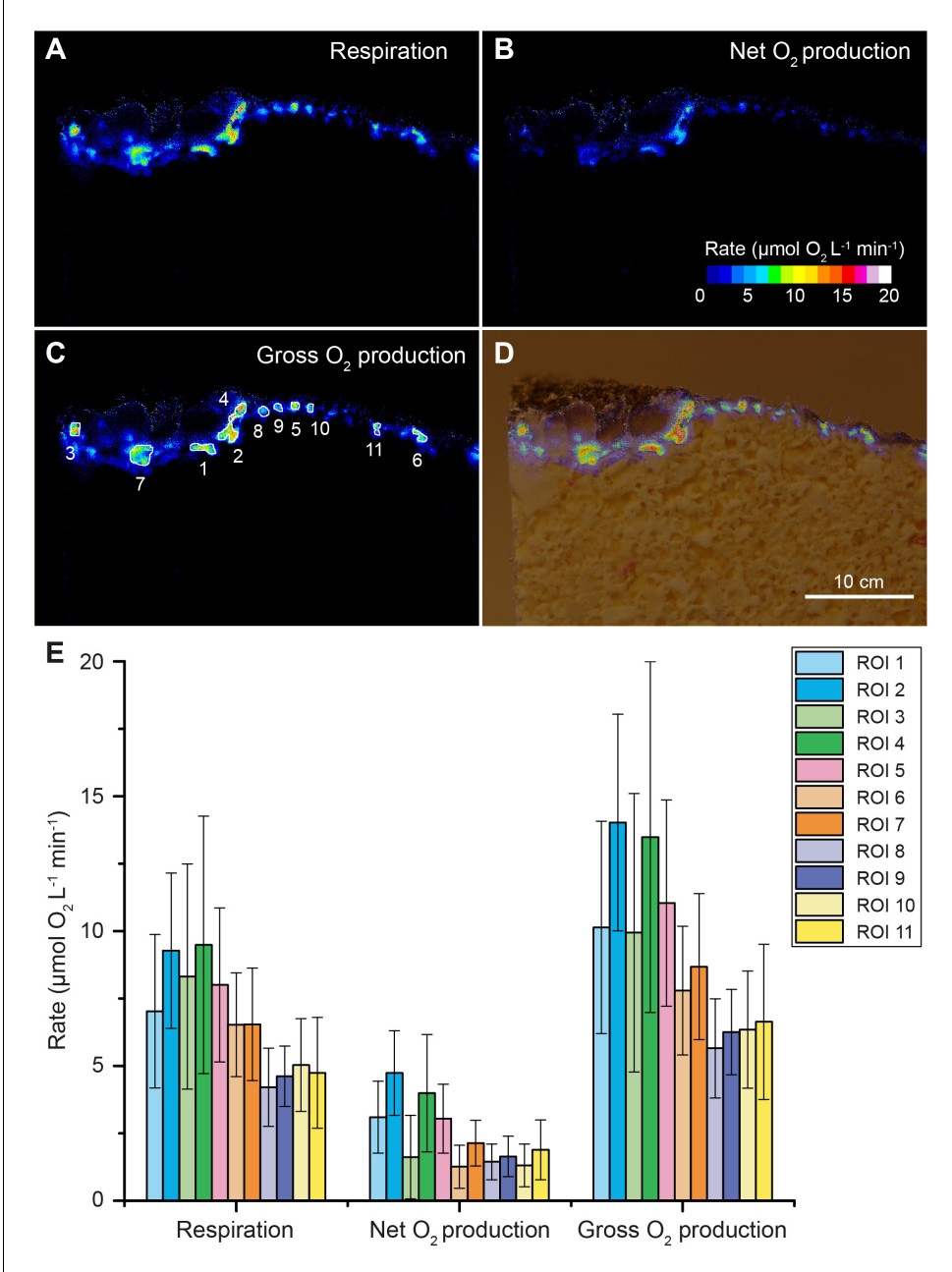

**Figure 2.** Oxygen consumption and NIR-driven oxygenic photosynthesis in beachrock. Cross-sectional images of initial $O_2$ consumption after onset of darkness (**A**) and maximum net photosynthetic $O_2$ production (**B**) after onset of actinic NIR illumination (740 nm, 28 μmol photons $m^{-2}$ $s^{-1}$) of the beachrock cross-section shown in *Figure 1*. (**C**) The NIR-driven gross photosynthesis was estimated by summing the absolute rates of net photosynthesis under NIR and $O_2$ consumption in the dark. (**D**) Overlay of gross photosynthesis distribution over a structural image of the beachrock cross-section. (**E**) Data for $O_2$ consumption, and NIR-driven net and gross photosynthesis were extracted for 11 regions of interest (ROI) in panel C and are presented as means ± standard deviation within the ROI.

in a thin agar layer adjacent to the actual phototrophic cells, which will underestimate the true dynamics (see Materials and methods), and our measurements were conducted at low NIR levels that may not represent saturated photosynthesis rates. Obviously, the realized total daily primary productivity will also be strongly modulated by the actual biomass distribution, porosity and diel light exposure within the beachrock, as well as the pronounced diel cycles of environmental

conditions on the beachrock platform (*Petrou et al., 2014*; *Schreiber et al., 2002*). Nevertheless, we argue that our experimental data point to a significant role of NIR-driven oxygenic photosynthesis in beachrock and potentially other endolithic habitats.

Based on the origin of isolates shown to employ FaRLiP (*Supplementary file 1*), it has been speculated that shaded soils, caves, plant canopies and thermal springs may be prime terrestrial habitats for cyanobacteria with Chl *f* (*Gan et al., 2015*; *Zhang et al., 2019*). Beachrock is widespread in intertidal zones on a global scale (*Vousdoukas et al., 2007*) and our study shows that these habitats may present a major ecological niche for cyanobacteria with FaRLiP and NIR-driven oxygenic photosynthesis in marine intertidal habitats. While we present the hitherto most detailed insight into the distribution and in situ activity of Chl *f*-containing cyanobacteria in intact samples from a natural habitat, there is now a need for more precise characterization of the microenvironment and metabolic activities of microorganisms in beachrock to assess the quantitative importance of NIR-driven oxygenic photosynthesis for system productivity.

The present study and the few other studies of natural habitats (*Supplementary file 1*) demonstrate that a key trait of such cyanobacteria is the formation of biofilms in strongly shaded environments below other algae, cyanobacteria or terrestrial plants, or in the twilight zone of caves (*Behrendt et al., 2019*). It remains to be explored to what extent the presence of cyanobacteria with FaRLiP-capability or constitutive high Chl *d* levels play a role for overall photosynthetic productivity in such habitats. That is, does the complementary photopigmentation of cyanobacteria with far-red shifted photopigments in the 'understory' of other oxygenic phototrophs lead to higher photosynthetic efficiency and productivity at the community/system level, for example in analogy to plant canopies? Such studies are complicated due to the compacted and often stratified structure of the natural communities, but the novel combination of structural and chemical imaging presented here seems a promising toolset for unraveling the ecological importance of FaRLiP and cyanobacteria with far red-shifted photopigments.

# Materials and methods

### Key resources table

| Reagent type (species) or resource | Designation | Source | Identifiers | Additional information |
|---|---|---|---|---|
| Chemical compound, drug | Agarose | Thermo Fisher | 16520100 | |
| Chemical compound, drug | PtTFPP | Frontier Scientific www.frontiersci.com/ | PtT975 | $O_2$-sensitive dye in the $O_2$ sensor nanoparticles (*Koren et al., 2015*) |
| Chemical compound, drug | Macrolex fluorescent yellow | Kremer Pigments www.kremer-pigmente.com/en/ | | Reference dye in the $O_2$ sensor nanoparticles (*Koren et al., 2015*) |
| Chemical compound, drug | PSMA (XIRAN) | Polyscope www.polyscope.eu/ | | Polymer used in the $O_2$ sensor nanoparticles (*Koren et al., 2015*) |
| Software, algorithm | Image J | http://rsb.info.nih.gov/ij/ | RRID:SCR_003070 | Used for calculations of $O_2$ concentration images and visualization of hyperspectral images, structure and $O_2$ concentration |
| Software, algorithm | Ratio Plus Image J plugin | http://rsb.info.nih.gov/ij/plugins/ratio-plus.html | | Used for calculations of image ratios |
| Software, algorithm | look@RGB | http://imaging.fish-n-chips.de | | Used for camera and LED control during image acquisition (*Larsen et al., 2011*) |
| Software, algorithm | Hyperspectral Imager V. 4.2 | PhiLumina, LLC, Gulfport, MS, USA www.philumina.com/ | | Used for hyperspectral image stack acquisition |

*Continued on next page*

*Continued*

| Reagent type (species) or resource | Designation | Source | Identifiers | Additional information |
|---|---|---|---|---|
| Software, algorithm | ENVI | L3 Harris Geospatial, Brromfield, CO, USA www.harrisgeospatial.com/ | | Used for hyperspectral image stack conversion |
| Software, algorithm | Look@MOSI | www.microsenwiki.net/ doku.php/hsimaging: hs_iman_howto | | Used for hyperspectral image analysis |
| Chemical compound, drug | Acetone Methanol Ammonium acetate Acetonitrile Ethylacetate | Sigma Aldrich | 650501 34860 543834 114291 103649 | Solvents used in HPLC analysis |
| Software, algorithm | OpenLAB CDS ChemStation Edition | Agilent Technologies | | Used for HPLC analysis |
| Other | Supor-200 polyethersulfone membrane disc filters (47 mm diameter, 0.2 μm pore size) | PALL | 63025 | Filters for seawater filtration |
| Commercial assay or kit | DNeasy PowerLyzer PowerSoil kit | QIAGEN | 12855 | Extraction of DNA from beachrock |
| Commercial assay or kit | DNeasy Power Water kit | QIAGEN | 14900 | Extraction of DNA from seawater |
| Commercial assay or kit | PCR reaction | PCRBIO | PB10.41–02 | Amplification of 16S rRNA gene |
| Sequence-based reagent | V3 | Eurofins | 5′-CCTAYGGGRBGCASCAG-3′ | PCR primer for 16S rRNA gene |
| Sequence-based reagent | V4 | Eurofins | 5′-GGACTACHVGGGTWTCTAAT-3′ | PCR primer for 16S rRNA gene |
| Software, algorithm | BLAST | NCBI http://blast.ncbi.nlm. nih.gov/Blast.cgi | RRID: SCR_004870 | |

## Field site and beachrock sampling

Beachrock fragments (5–10 cm) from the upper black-brown zone of the intertidal beachrock platform on Heron Island (Great Barrier Reef, Queensland, Australia; 23°26.5540S, 151°54.7420E); the field site is described in detail elsewhere (*Cribb, 1966*; *Díez et al., 2007*; *Trampe and Kühl, 2016*). Beachrock was sampled at low tide and cut into smaller subsamples with smooth vertical cross sections (ca. $2 \times 2 \times 1$ cm$^3$) using a seawater-flushed stone saw under dim light. These samples were subsequently stored and transported dry and dark prior to the imaging experiments at University of Technology Sydney, which commenced within a few days after sampling. Other black beachrock samples were either processed at the Heron Island Research Station for DNA extraction, or were frozen in liquid $N_2$ on site and shipped on dry ice to Denmark for subsequent pigment extraction and HPLC analysis.

## Hyperspectral imaging

### Imaging setup

Hyperspectral reflectance imaging was done on vertical cross-sections of beachrock as previously described *Trampe and Kühl (2016)*. The samples were submerged in seawater (23°C and salinity = 35) with the cross-section facing the objective of a dissection zoom microscope (Leica, Germany) with a hyperspectral camera system (100T-VNIR; Themis Vision Systems, St. Louis, MO) (*Kühl and Polerecky, 2008*) connected via the C-mount video adapter. A fiber-optic halogen lamp with an annular ring-light (KL-2500 and Annular Ring-light; Schott AG, Mainz, Germany) mounted on the objective of the dissection microscope was used as a light source for the hyperspectral image acquisition. Additional hyperspectral measurements of reflected light were performed (at similar

distance and microscope settings as used for the beachrock samples) on a calibrated 20% reflectance standard (Spectralon SRM-20; LabSphere Inc, North Sutton, NH, USA).

## Hyperspectral image analysis

Using the manufacturers software, (PhiLumina Hyperspectral Imager V. 4.2; PhiLumina, LLC, Gulfport, MS), dark-corrected, hyperspectral image stacks of beachrock reflection were converted to hyperspectral reflectance images (in units of % reflectance) by normalization to dark-corrected reflection stacks recorded using the reflectance standard. Subsequent file format conversion, and image cropping were performed in ENVI (Exelis Visual Information Solutions, Boulder, CO), before further image processing with the software Look@MOSI (freeware available at www.microsen-wiki. net/doku.php/hsimaging:hs_iman_howto).

RGB images were constructed from the reflectance measurements at 650 nm (Red), 550 nm (Green), and 450 nm (Blue), as extracted from the calibrated hyperspectral image stacks. The fourth derivative of the hyperspectral reflectance stacks was calculated using the Look@MOSI software, yielding the relative extent of light attenuation at wavelengths representative for Chl *a* (670–680 nm) and Chl *f* (718–722 nm) absorption. The resulting grayscale images were then used for construction of false-color-coded images, showing the spatial coverage of the two defined spectral signatures. The extraction of spectral information from areas of interest (covering hotspots of Chl *f*) was performed as previously described (*Polerecky et al., 2009*). The resulting images were first cropped to be of computational sizes for the Look@MOSI software, and were stitched back together in Photoshop CC 2015.1.2 (Adobe Systems Incorporated, San Jose, CA) after computation. The relative absorbance/abundance of Chl *f* was quantified by scaling of the relative extent of light attenuation as calculated above for Chl *f*. Pixel intensity values from the earlier calculated gray scale images were assigned a scale ranging from 0 to 1, and were finally displayed using a color scaled lookup table in ImageJ (http://rsbweb.nih.gov/ij/). This yielded false-color coded images with values between 0 and 1, displaying a relative measure of Chl *f* abundance according to the amount of light attenuation obtained from Chl *f* absorption.

## Pigment analysis

Two vertically separated layers from the black beachrock zone were analyzed for composition of photopigments, that is in the top layer (0–2 mm) and in a more distinct green deeper layer (2–5 mm). Samples from the two layers were kept at −80°C until further analysis by high-pressure liquid chromatography (HPLC) as described in detail elsewhere (*Trampe and Kühl, 2016*). Homogenized beachrock samples were transferred to 1.5 mL Eppendorf tubes for pigment extraction. The pigments were extracted by adding 0.8 mL acetone:methanol (7:2 vol:vol) to each tube, which was briefly vortexed, and then kept on ice for a 2 min extraction time in darkness. Subsequently, samples were sonicated in an ice-cooled high-power ultrasonicating bath (Misonix 4000; Qsonica LLC., Newtown, CT) in darkness for 10 s consisting of 10 pulses of 1 s ON and 1 s OFF (with an amplitude setting of 100%), and were then centrifuged at ~12,000 g for 1 min in a mini centrifuge (MiniSpin, Eppendorf AG, Hamburg, Germany). The supernatant was filtered through a 0.45 µm pore size syringe filter (Satorius Minisart SRP 4 filter; Sartorius AG, Goettingen, Germany), and 200 µL filtrate was then mixed with 15 µL ammonium acetate (1 M) in 0.3 mL HPLC vials. 100 µL of the extract was then immediately injected into the HPLC. Pigment extracts were separated and analyzed in the HPLC by a diode array detector (HPLC-DAD and Agilent 1260 Infinity; Agilent Technologies, Santa Clara, CA) fitted with Ascentis C18 column (25 cm × 4.6 mm, Sigma-Aldrich cat. no. 581325 U), detecting specific absorption wavelengths of compounds. The extracts were run at a constant column temperature of 30°C for 69 min., and a flow-rate of 1.0 mL min$^{-1}$ in a changing gradient of solvent A (methanol:acetonitrile:water, 42:33:25, vol/vol/vol), and solvent B (methanol:acetonitrile: ethylacetate, 50:20:30, vol/vol/vol), where the mobile phase changed linearly from 30% solvent B at the time of injection to 100% at 52 min, staying at 100% for15 min before returning to 30% within 2 min. Spectral comparisons from *Chen and Blankenship (2011)*, and *Trampe and Kühl (2016)* were used for identification of Chl *a*, Chl *d*, and Chl *f* from the HPLC chromatograms. Elution profiles from the absorbance detector signal at 430 nm were used to calculate pigment ratios from the derived integrated peak areas for each of the identified pigments of interest, using the manufacturer's software (OpenLAB CDS ChemStation Edition; Agilent Technologies).

## Amplicon sequencing

A survey of the microbial communities in the beachrock was performed based on high-throughput sequencing of the 16S rRNA community gene pool, of which details will be published elsewhere. In brief, genomic DNA was isolated from portions (0.1–0.4 g) of freshly sampled black beachrock using the DNeasy PowerLyzer PowerSoil kit (QIAGEN, Germany). Costal seawater was sampled by filtration of 2.3 L seawater onto Supor-200 polyethersulfone membrane disc filters (47 mm diameter, 0.2 µm pore size; PALL cat. no. 63025) and genomic DNA was isolated using the DNeasy PowerWater kit (QIAGEN, Germany). This genomic DNA was used as template for PCR amplification with primers targeting the V3 and V4 regions of the bacterial/chloroplast 16S rRNA gene. The PCR products were sequenced with the Illumina MiSeq platform and the sequence data analyzed with the Qiime2 pipeline at the Section for Microbiology, Department of Biology, University of Copenhagen.

## Experimental setup for $O_2$ imaging

The experiments were performed in a small glass aquarium ($15 \times 10 \times 15$ cm$^3$) filled with filtered seawater (salinity 35). A lid with two inlets was placed on the aquarium. One inlet was used as a gas inlet, the second inlet was used to insert a robust fiber-optic $O_2$ sensor (OXR430 connected to Fire-Sting GO$_2$ meter; PyroScience GmbH, Aachen, Germany) for monitoring the $O_2$ concentration in the bulk seawater. The beachrock sample was mounted with a smooth vertical cross-section parallel to the aquarium glass wall. The camera used for $O_2$ imaging and the excitation LED were placed perpendicular to the sample cross-section. Actinic NIR illumination was provided by a 740 nm LED (LZ4-40R300; LED Engin, Inc, San Jose, CA; HBW 25 nm) providing a NIR photon irradiance (integrated over 715–765 nm) of 28 µmol photons m$^{-2}$ s$^{-1}$, as measured with a calibrated spectroradiometer (MSC15, GigaHertz-Optik GmbH, Germany).

A beachrock sample painted on one side with the $O_2$ sensor nanoparticle paint (see below and *Figure 1—figure supplement 6*) was placed into the aquarium, and the surrounding seawater was flushed with pure $N_2$ for at least 30 min to completely remove $O_2$, as confirmed by the fiber-optic $O_2$ sensor. After anoxic conditions were reached, measurements of the change in $O_2$ concentration over the beachrock cross section in darkness and under NIR illumination were performed under stagnant anoxic conditions in the surrounding seawater (*Figure 1C*, *Figure 1—figure supplement 7*). Images were recorded at 5 min intervals relative to onset of actinic light or darkness. This enabled both recording of steady-state $O_2$ images in light and darkness (=homogeneous anoxia), as well as the dynamic change in $O_2$ distribution upon NIR irradiation. Proxies for the NIR-driven net photosynthesis, $P_N$, and apparent dark respiration, $R_D$, were calculated by subtracting images taken with a 5 min interval just before and after onset or eclipse of the actinic NIR light, respectively. Gross photosynthesis was estimated as $P_G = P_N+|R_D|$. To avoid any interference from surrounding light reaching the sample, the entire setup was covered thoroughly with black fabric. All measurements were performed at room temperature ($23 \pm 1$°C).

It should be noted that the $O_2$ measurements were done over minute intervals in the thin agar layer with $O_2$-sensitive nanoparticles coating the beachrock with the photosynthetic cells, where the observed $O_2$ levels are affected both by cell activity and diffusional exchange with the surroundings. This leads to some diffusive smearing, and the observed dynamics in $O_2$ and thus the derived reaction rates likely represent an underestimate of 'true' reaction rates (*Santner et al., 2015*).

## Chemical imaging of $O_2$
### Imaging system and application of $O_2$ sensor nanoparticles
We used recently published protocols for imaging $O_2$ concentration over complex bioactive surfaces by coating beachrock cross-sections with $O_2$-sensitive luminescent sensor nanoparticles (*Koren et al., 2015*; *Koren et al., 2016*) followed by ratiometric luminescence imaging of the coated surface with a DSLR camera (without NIR filter and with a 530 nm long pass filter mounted on the camera objective) during brief excitation with blue LED light (445 nm). Details of the imaging system and image acquisition software (*Larsen et al., 2011*), nanoparticle fabrication (*Koren et al., 2015*; *Moßhammer et al., 2019*) and biocompatibility (*Trampe et al., 2018*) are given elsewhere.

## Optical O$_2$ sensor nanoparticle paint

First, we attempted to spray-paint beachrock sections with sensor nanoparticles using a paintbrush according to *Koren et al. (2016)*, but the porous structure of the beachrock prevented saturation and a homogenous coating of cross-sections. Instead, we coated beachrock cross-sections (as well as glass slides used for calibration) with a thin (<1 mm thick) layer of agarose containing O$_2$ sensor nanoparticles, inspired by earlier work on seagrass rhizosphere O$_2$ imaging (*Koren et al., 2015*). For this, 40 mg of UltraPure Low Melting Point Agarose (16520100; https://www.thermofisher.com/) was melted in 2 mL filtered seawater, which was then kept at ~35℃ and mixed with 2 mL of a pre-warmed (~35℃) O$_2$ sensor nanoparticle solution (2.5 mg mL$^{-1}$). This 'sensor paint' was then applied on a vertical cross-section of a beachrock sample (or a glass microscope slide) using a small paint brush. After solidification of the sensor paint, the coated beachrock was transferred into the aquarium and left there prior to experiments to allow acclimatization. Throughout sample preparation, exposure to high light levels was avoided. This procedure produced a stable homogenous coating of beachrock cross-sections (*Figure 1—figure supplement 6*) with an O$_2$ sensing layer that could be easily peeled off, enabling the same sample to be used for subsequent CLSM or hyperspectral imaging.

## Imaging and image analysis

The O$_2$-dependend red emission, and the constant green reference emission from the sensor nano-particles in the paint during brief excitation pulses from a blue LED (445 nm) were recorded in RGB pictures with a DSLR camera system (*Larsen et al., 2011*) imaging coated beachrock sections or coated microscope glass slides. Mapping of NIR-driven oxygenic photosynthesis was done by O$_2$-imaging of beachrock biofilm cross-sections immersed in anoxic (N$_2$ flushed) seawater under illumination with a NIR LED. Acquired RGB images were split into red, green, and blue channels and analyzed using the freely available software ImageJ (http://rsbweb.nih.gov/ij/; RRID:SCR_003070). First, the red channel images (recording the O$_2$-sensitive emission of the sensor nanoparticles) and the green channel images (recording the constant emission of a reference dye in the nanoparticles) were divided using the ImageJ plugin Ratio Plus (http://rsb.info.nih.gov/ij/plugins/ratio-plus.html) in order to get ratio images, R (=red channel/green channel).

Background fluorescence from Chl *a* (at >680 nm) excited by the excitation light (445 nm) during O$_2$ imaging can potentially overlap with the red channel. Furthermore, such Chl *a*-based photosynthesis may also generate a small amount of O$_2$ during the brief excitation pulse. To account for such potential artefacts, and as we aimed to visualize changes in O$_2$ concentration (ΔO$_2$) attributed to Chl *f*-based photosynthesis, all ratio images were subtracted from ratio images recorded after 45 min in the dark in anoxic water, R$_{dark}$, (ΔR = R$_{dark}$ R). Further, to avoid any cross-talk of NIR into the red channel, O$_2$ images were always recorded during a brief (<1 s) darkening during image acquisition.

## Calibration

For calibration, a glass slide was coated with a thin layer of the O$_2$ sensor nanoparticle paint. The coated slide was placed in the experimental setup with aerated seawater and imaged at identical camera settings as used for beachrock sample imaging. Subsequently, the O$_2$ content of the seawater was decreased by flushing it intermittently with N$_2$ gas under constant monitoring by a fiber-optic O$_2$ microsensor (OXR430 connected to FireSting GO$_2$ meter; PyroScience GmbH, Aachen, Germany). Calibration curves were obtained from RGB images of the calibration target recorded under a series of known seawater O$_2$ concentrations (ranging from 100% air saturation to anoxia) using a ROI covering the whole field of view. Plotting R versus O$_2$ showed an exponential decay with increasing O$_2$ concentration (*Figure 1—figure supplement 8*), as commonly observed for optical O$_2$-sensing materials (*Koren and Kühl, 2018*; *Moßhammer et al., 2019*). To enable calibration of background corrected experimental ratio images (see above), we generated a calibration curve by plotting ΔR (=R$_{anoxic}$ R) versus O$_2$ concentration (*Figure 1—figure supplement 8B*), which was fitted with an exponential function in Image J. The ΔR images from the experiments were then calibrated in Image J using the exponential fit in the calibration function.

## Confocal laser scanning microscopy

After the $O_2$-imaging, the nanoparticle paint was peeled off the beachrock sections, and a smaller subsection of the flat beachrock cross section (previously used for hyperspectral or chemical imaging) was imaged at 200x and 400x magnification on an inverted confocal laser scanning microscope (Nikon A1R, Japan) with the beachrock cross section facing downwards in a 35 mm coverslip glass bottom petri dish (World Precision Instruments). Care was taken to keep the beachrock cross sections intact and oriented to enable identification and alignment of CLSM data to areas of interest exhibiting NIR-driven $O_2$ production.

The microscope was equipped with a motorized xyz sample holder and was able to acquire hyperspectral fluorescence image stacks over a large sample area by sequential scanning, with subsequent automatic stitching of the acquired images in the microscope software (NIS elements AR, Nikon, Japan). The sample was excited by 488 nm laser light (laser power 1.2 mW, 0.5 frames per seconds) and spectral fluorescence (500–750 nm) was acquired at 10 nm resolution by the spectral PMT-array detector on the CLSM microscope. First, the beachrock cross-section was imaged at 200x magnification obtaining hyperspectral image stacks of 5 confocal layers (step size 5 µm) for each field of view. Scanned images were stitched to make a 4 mm x 11 mm large image and were then de-convolved (numerical aperture of 0.75, pinhole size 177.52, refractive index 1 for 26 emission channels 500–750 nm) using the NIS elements AR software (Version 4.60, Nikon, Japan). Subsequently, a more detailed scan of the same cross sectional area was done at 400x magnification (laser power 1.2 mW, 0.063 frames per seconds), and spectral fluorescence (500–750 nm) was acquired at 10 nm resolution by the spectral PMT-array detector on the CLSM microscope obtaining hyperspectral fluorescence images in one z- plane.

We note that the beachrock cross-section was not perfectly smooth and single plane recording at 400x led to less signal in parts of the cross-section, where the surface was out of focus. Nevertheless, we could still identify most of the hot spots recorded at 200x magnification and resolve the shape of individual cells and cell clusters. Scanned images were stitched to make a 3.5 mm x 5.2 mm large image and were then de-convolved (numerical aperture of 1.00, pinhole size 177.52, refractive index 1.515 for 26 emission channels 500–750 nm). Regions of interest (ROI) were selected from the obtained cross-sectional scans and spectral fluorescence features were captured and false-color coded for particular regions of interest. CLSM scans were false-color coded to highlight the fluorescence of Chl *a* (690–700 nm), phycobiliproteins (650–660 nm), and Chl *f* (740–750 nm; cf. *Majumder et al., 2017*).

## Acknowledgements

We acknowledge the excellent technical assistance in the field and during laboratory measurements by Sofie L Jakobsen, Veronica M Petersen, and the staff at Heron Island Research Station. Lorrie Maccario is thanked for assistance with the MiSeq amplicon sequencing. Work at Heron Island was conducted under permit no. G16/38423.1 from the Great Barrier Reef Marine Parks authority

## Additional information

### Funding

| Funder | Grant reference number | Author |
| --- | --- | --- |
| Det Frie Forskningsråd | DFF-8021-00308B | Michael Kühl |
| Det Frie Forskningsråd | DFF-8022-00301B | Michael Kühl |
| Det Frie Forskningsråd | DFF-4184-00515B | Michael Kühl |
| Villum Fonden | 00023073 | Michael Kühl |
| Poul Due Jensen Fonden | | Klaus Koren |

The funders had no role in study design, data collection and interpretation, or the decision to submit the work for publication.

## Author contributions

Michael Kühl, Conceptualization, Resources, Data curation, Formal analysis, Supervision, Funding acquisition, Validation, Investigation, Visualization, Methodology, Project administration, Writing, Original draft, Review and editing; Erik Trampe, Conceptualization, Data curation, Formal analysis, Validation, Investigation, Visualization, Methodology, Review and editing; Maria Mosshammer, Investigation, Methodology, Review and editing; Michael Johnson, Resources, Software, Formal analysis, Investigation, Visualization, Methodology, Review and editing; Anthony WD Larkum, Investigation, Visualization, Methodology, Review and editing; Niels-Ulrik Frigaard, Conceptualisation, Resources, Data curation, Formal analysis, Validation, Investigation, Methodology, Review and editing; Klaus Koren, Conceptualization, Data curation, Software, Formal analysis, Validation, Investigation, Visualization, Methodology, Review and editing

## Author ORCIDs

Michael Kühl https://orcid.org/0000-0002-1792-4790
Erik Trampe https://orcid.org/0000-0003-3249-0297
Maria Mosshammer https://orcid.org/0000-0002-7296-8673
Niels-Ulrik Frigaard https://orcid.org/0000-0002-9389-8109
Klaus Koren https://orcid.org/0000-0002-7537-3114

## Decision letter and Author response

Decision letter https://doi.org/10.7554/eLife.50871.sa1
Author response https://doi.org/10.7554/eLife.50871.sa2

## Additional files

### Supplementary files

• Supplementary file 1. Overview of cyanobacterial strains and enrichments reported to contain Chl *f*.

• Supplementary file 2. List of most abundant OTUs related to oxygenic phototrophs in black beachrock and seawater.

• Transparent reporting form

### Data availability

All data generated or analysed during this study are included in the manuscript and supporting files.

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
