## [Decision Letter]

**Acceptance summary:**

Your work enhances our appreciation of microbial photosynthetic diversity. It demonstrates that recently studied Chl *f*-containing organisms (likely cyanobacteria) can be found in ecological niches that are red-shifted. This important result extends relatively recently acquired biochemical/biophysical observations into a natural context. This is an exciting finding because it expands the known cyanobacterial niche in nature, and may be relevant to future bioengineering efforts to optimize solar energy conversion. This work should be of interest anyone who cares about microbial metabolic diversity and ecology, including those seeking to harness microbial strategies for sustainable energy applications.

**Decision letter after peer review:**

Thank you for submitting your article "Substantial near-infrared radiation-driven photosynthesis of chlorophyll *f*-containing cyanobacteria in a natural habitat" for consideration by *eLife*. Your article has been reviewed by three peer reviewers, and the evaluation has been overseen by a Reviewing Editor and Christian Hardtke as the Senior Editor. The following individuals involved in review of your submission have agreed to reveal their identity: Chris Gisriel (Reviewer #3).

The reviewers have discussed the reviews with one another and the Reviewing Editor has drafted this decision to help you prepare a revised submission.

Summary:

This paper demonstrates that recently studied Chl *f*-containing organisms can be found in ecological niches that are red-shifted. This important result extends relatively recently acquired biochemical/biophysical observations into a natural context. Based on previous studies, and it is likely that these organisms are cyanobacteria. Assuming this is the case, this is an exciting finding because it expands the known cyanobacterial niche in nature, and potentially may be relevant to future bioengineering efforts to optimize solar energy conversion. The consensus of the reviewers is that the technical approaches in the paper are rigorous, but a few essential revisions are needed prior to publication.

Essential revisions:

1) The identity of Chl *f* must be verified unambiguously. It appears that pigment analyses were performed on identical samples in Trampe and Kuhl, 2016. At a minimum, the authors should revise the text to clearly site this reference so readers can find this data; but the inclusion of pigment analyses from these samples, as supplemental data, would be preferred.

2) The authors assert that the Chl *f*-containing organisms in their samples are cyanobacteria. While this seems highly likely, based on the literature, this study would nonetheless be significantly strengthened by inclusion of supporting data linking the measurements in these samples to organismal identity. To this end, please include 16S rDNA data. If there is a compelling reason why 16S rDNA data are unnecessary to link Chl *f* to cyanobacteria in these natural samples, please articulate this in the text.

3) Finally, the current manuscript is less accessible to a broad biology audience than it should be. Only at the end of the Results/Discussion section, do the authors explain why a general audience should find this interesting. These points are very good, and should be made earlier. Before diving into the substantive details (which should remain), please revise so as to motivate non-experts to keep reading by explaining the significance of this work in more general terms.

---

## [Author Response]

Essential revisions:1) The identity of Chl f must be verified unambiguously. It appears that pigment analyses were performed on identical samples in Trampe and Kuhl, 2016. At a minimum, the authors should revise the text to clearly site this reference so readers can find this data; but the inclusion of pigment analyses from these samples, as supplemental data, would be preferred.

We have now included supporting data that verify the identity of Chl *f* and other chlorophylls in the beachrock. We used HPLC-based pigment analysis (according to Trampe and Kühl, 2016) on black beachrock samples (taken from the same beachrock zone as the samples used for hyperspectral and O_2_ imaging). Analysis of chromatograms and absorption spectra of different eluent fractions showed the presence of Chl *a* and *f*, and minor amounts of Chl *d* in the black beachrock (both in the upper 2 mm and in a deeper green endolithic zone 2-5 mm below the beachrock surface), while other chlorophylls like Chl *b* and *c* (indicative of microalgae) were absent in the samples. The amount of Chl *f* relative to Chl *a* in these samples ranged from 3.5% in the mixed surface layer to 6% in the deeper green layer, while Trampe and Kühl, 2016, reported Chl *f* amounts ranging from 0.5 to 5% of Chl *a* in beachrock.

Changes in the revised manuscript:

New text was added in the Results and Discussion section (paragraph one) and in the Materials and methods section of the revised manuscript (subsection “Pigment analysis”), and we added new supporting data (Figure 1—figure supplement 3) showing chromatograms and the absorption spectra of the Chl *f* fraction in the pigment extracts from the beachrock.

2) The authors assert that the Chl f-containing organisms in their samples are cyanobacteria. While this seems highly likely, based on the literature, this study would nonetheless be significantly strengthened by inclusion of supporting data linking the measurements in these samples to organismal identity. To this end, please include 16S rDNA data. If there is a compelling reason why 16S rDNA data are unnecessary to link Chl f to cyanobacteria in these natural samples, please articulate this in the text.

We have now included supporting data on the identity of oxygenic phototrophs in the black beachrock. We performed amplicon sequencing of the 16S rRNA gene pool in the upper 2 mm and in a deeper green endolithic zone 2-5 mm below the surface of black beachrock collected from the same beachrock zone as the samples used for hyperspectral and O_2_ imaging and pigment analysis. These analyses showed no presence of OTUs related to microalgal chloroplasts in the black beachrock, while amplicon sequencing of the local seawater revealed a substantial fraction of OTUs that clustered with algal chloroplasts (Supplementary file 2). Hence, our NIR-driven O_2_ production in the beachrock could be assigned to the presence of cyanobacteria. Interestingly, several of the predominant OTUs were related to cyanoabacteria capable of far-red light photoacclimation (FaRLiP) and synthesis of Chl *f*. However, we note that a detailed investigation of the microbial diversity in beachrock was beyond the scope of the present study and a more comprehensive study based on both metagenomics and 16S amplicon sequencing is currently underway.

Changes in the revised manuscript:

New text was added in the Results and Discussion section (paragraph two) and in the Materials and methods section (subsection “Amplicon sequencing”) of the revised manuscript, and we added new supporting data (Supplementary file 2) listing the most prominent OTUs in the beachrock and seawater samples relating to cyanobacteria and microalgal chloroplasts.

3) Finally, the current manuscript is less accessible to a broad biology audience than it should be. Only at the end of the Results/Discussion section, do the authors explain why a general audience should find this interesting. These points are very good, and should be make earlier. Before diving into the substantive details (which should remain), please revise so as to motivate non-experts to keep reading by explaining the significance of this work in more general terms.

Thank you for this suggestion. We have now explained the significance of the work in more detail in the introductory paragraphs.

Changes in the revised manuscript:

We have moved (and slightly reformulated) a paragraph from the Results and Discussion section to the Introduction (paragraph three) stating relevance of this work for understanding the actual role of cyanobacteria with far-red shifted photopigments in nature and their potential application in an applied context.